# Antifungal and Anti-Inflammatory Potential of *Bupleurum rigidum* subsp. *paniculatum* (Brot.) H.Wolff Essential Oil

**DOI:** 10.3390/antibiotics10050592

**Published:** 2021-05-17

**Authors:** Mónica Zuzarte, Pedro M. P. Correia, Jorge M. Alves-Silva, Maria J. Gonçalves, Carlos Cavaleiro, Teresa Cruz, Lígia Salgueiro

**Affiliations:** 1Faculty of Medicine, Coimbra Institute for Clinical and Biomedical Research (iCBR), University of Coimbra, 3000-548 Coimbra, Portugal; jmasilva@student.ff.uc.pt; 2Center for Innovative Biomedicine and Biotechnology (CIBB), University of Coimbra, 3000-548 Coimbra, Portugal; 3Clinical Academic Centre of Coimbra (CACC), 3000-548 Coimbra, Portugal; 4Faculty of Sciences, BioISI—Biosystems & Integrative Sciences Institute, University of Lisboa, 1749-016 Lisboa, Portugal; pmpcorreia@fc.ul.pt; 5Faculty of Pharmacy, University of Coimbra, 3000-548 Coimbra, Portugal; mpinho@ff.uc.pt (M.J.G.); cavaleir@ff.uc.pt (C.C.); trosete@ff.uc.pt (T.C.); ligia@ff.uc.pt (L.S.); 6Chemical Process Engineering and Forest Products Research Centre (CIEPQPF), Department of Chemical Engineering, University of Coimbra, 3030-790 Coimbra, Portugal; 7Centre for Neuroscience and Cell Biology (CNC), 3000-548 Coimbra, Portugal

**Keywords:** antifungal, germ tube, biofilm, anti-inflammatory

## Abstract

Fungal infections remain a major health concern with aromatic plants and their metabolites standing out as promising antifungal agents. The present study aims to assess, for the first time, the antifungal and anti-inflammatory potential of *Bupleurum* subsp. *paniculatum* (Brot.) H.Wolff essential oil from Portugal. The oil obtained by hydrodistillation and characterized by GC-MS, showed high amounts of monoterpene hydrocarbons, namely α-pinene (29.0–36.0%), β–pinene (26.1–30.7%) and limonene (10.5–13.5%). The antifungal potential was assessed, according to CLSI guidelines, against several clinical and collection strains. The essential oil showed a broad fungicidal effect being more potent against *Cryptococcus neoformans* and dermatophytes. Moreover, a significant germ tube inhibition was observed in *Candida albicans* as well as a disruption of mature biofilms, thus pointing out an effect of the oil against relevant virulent factors. Furthermore, fungal ultrastructural modifications were detected through transmission electron microscopy, highlighting the nefarious effect of the oil. Of relevance, the oil also evidenced anti-inflammatory activity through nitric oxide inhibition in macrophages activated with lipopolysaccharide. In addition, the essential oil’s bioactive concentrations did not present toxicity towards macrophages. Overall, the present study confirmed the bioactive potential of *B. rigidum* subsp. *paniculatum* essential oil, thus paving the way for the development of effective drugs presenting concomitantly antifungal and anti-inflammatory properties.

## 1. Introduction

Fungal infections account for nearly 1.6 million deaths worldwide, thus presenting a huge socio-economic burden, frequently neglected by public health authorities [1]. Most fungal infections affect mucous membranes, especially vaginal and mouth membranes [2] and, despite being primarily superficial, mycosis can rapidly evolve into systemic infections, especially in immunocompromised individuals. These type of infections are mainly caused by *Candida albicans*, nevertheless other *Candida* species are becoming important etiological agents such as *C. krusei*, *C. parapsilosis* and *C. tropicalis* [3]. Several features contribute to *Candida* spp. virulence including yeast-to-hypha morphological transition and biofilm formation [4]. The first, starts with germ tube formation that develop into hypha promoting tissue penetration and immune avoidance, thus resulting in a more effective infection. In addition, this transition is necessary for biofilm formation [5]. These complex structures can be formed in both biological and artificial surfaces and constitute a major clinical burden due to their higher resistance to antifungal drugs [6]. In fact, for most conventional antifungals, the minimal inhibitory concentration necessary to inhibit biofilms is 100-fold higher than the concentrations needed to inhibit planktonic cells [7]. Fungal infections can also be caused by other yeast such as *Cryptococcus neoformans* which affects the central nervous system and by dermatophytes of the *Trichophyton*, *Microsporum* and *Epidermophyton* genera. Dermatophytes are able to colonize keratinized structures, such as hair, skin and nails. Furthermore, filamentous fungi from the *Aspergillus* genus, especially *A. fumigatus*, *A. flavus* and *A. niger*, are also important agents in the development of bloodstream systemic infections as well as pulmonary and allergic infections [8].

Current antifungal therapy continues to present several limitations due to safety issues, pharmacokinetics, narrow activity spectrum and low number of targets [9] For example, polyenes and amphotericin B are highly toxic whereas azoles, like fluconazole, only present a fungistatic effect. In addition, the emergence of resistant strains also compromises the effectiveness of these antifungals [10,11].

Importantly, fungal infections also trigger an inflammatory response when fungal epitopes such as zymosan, bind to toll-like receptors (TLRs) in the host cells. TLRs binding triggers a series of intracellular cascades that culminate with the activation of pro-inflammatory transcription factors as well as the expression of several inflammatory cytokines such as interleukin-1β (IL-1 β), IL-6 and TNF-α, which further fuel the inflammatory response [12,13]. In addition, TLRs binding also leads to the production of reactive oxygen and nitrogen species, such as nitric oxide [14] due to an increase in the production of key pro-inflammatory enzymes, such as inducible nitric oxide synthase (iNOS) and cyclooxygenase-2 (COX-2) [15].

In some disorders such as gastrointestinal tract inflammatory diseases, like Chron’s disease, the colonization by *Candida* spp. is promoted leading to an exacerbated production of inflammatory cytokines which further worsens the inflammatory condition [16]. This triggers an endless circle of infection-inflammation that is generally controlled by the use of anti-inflammatory drugs. Nevertheless, these drugs have several side effects, including gastrointestinal, cardiovascular, hepatic, renal, cerebral and pulmonary complications [17]. Therefore, the development of new antifungal and anti-inflammatory agents with less side effects is mandatory and compounds showing concomitantly both properties represent a valuable therapeutic strategy.

In this context, aromatic and medicinal plants, especially those belonging to the Lamiaceae and Apiaceae families, emerge as promising sources of antifungal and/or anti-inflammatory agents. Indeed, species from the Apiaceae family have been used for centuries in traditional medicine for the treatment of several pathologies. Their beneficial effects are often related to the presence of essential oils, with reported antifungal and anti-inflammatory properties [18,19]. However, many uses are based on empiric knowledge, and the spectrum of action as well as the underlying mechanisms of action remain unknown, thus justifying scientific-based studies to further contribute to the development of effective therapeutics.

*Bupleurum* L. is one of the largest genus of the Apiaceae (Umbelliferae) family. Species from this genus are used in Asia, North Africa and Europe for their medicinal properties and have been used for millennia in traditional Chinese medicine [20]. For example, in Spain these species are used for the treatment of topical inflammation, as anti-infective and to promote wound healing [21], thus suggesting an antimicrobial and anti-inflammatory potential. Several secondary metabolites have also been described in *Bupleurum* species, including saikosaponins, flavonoids and volatile oils [20]. Moreover, different studies have also shown their anti-inflammatory [22,23,24]; hepatoprotection [25,26], antioxidant [22,27,28], antifungal and antibacterial potential [29,30]. In addition, a patent for the use of *B. fruticosum* essential oil as an anti-ageing agent has been filled by L’Oréal [31], thus reinforcing the industrial and pharmaceutical interest of this genus.

Among *Bupleurum* species, *B. rigidum* L. is one of the most widely represented in Portugal. This species is widely used in Spain for the treatment of topical inflammation, as anti-diarrheic and against furuncles [21]. Two subspecies of *B. rigidum* have been described, namely subsp. *paniculatum* and subsp. *rigidum*. *B. rigidum* subsp. *paniculatum* (Brot.) H.Wolff is an endemic species to the south and center of Portugal, Andalucía (Spain) and North Africa [32] and, up to date, no studies on this subspecies have been conducted. Therefore, and as part of our ongoing studies on the valorization of wild Apiaceae species, we aim to characterize the essential oil composition of *B. rigidum* subsp. *paniculatum* from Portugal and assess its bioactive potential. A combined antifungal and anti-inflammatory potential for *B. rigidum* subsp. *paniculatum* essential oil is reported, with emphasis on its effect against *C. albicans* virulence factors.

## 2. Results

### 2.1. Essential Oil Composition

Table 1 summarizes the chemical profile of two samples of *B. rigidum* subsp. *paniculatum* analyzed by GC-MS. A yield of 0.7% (*w*/*v*) was obtained for the essential oil from Coimbra and 0.6% for the oil from Fátima. More than 95% of the compounds were identified in both samples.

The chemical analysis showed that both essential oils are characterized by the presence of high amounts of monoterpene hydrocarbons (94.4–96.6%), mainly α-pinene (29.0–36.0%), β-pinene (26.1–30.7%) and limonene (10.5–13.5%). The essential oil from Coimbra region represents a much larger population and, therefore, was selected to assess the antifungal and anti-inflammatory potential of the oil.

### 2.2. Antifungal Activity

#### 2.2.1. The Essential Oil Showed a Broad-Spectrum Fungicidal Effect

The antifungal potential of *B. rigidum* subsp. *paniculatum* essential oil was assessed against several yeast, dermatophyte and *Aspergillus* strains as depicted in Table 2. Besides collection strains, fungal isolates obtained from the clinic were also assessed, thus providing data on the antifungal potential of the oil against more resistant strains. Overall, the essential oil showed a significant antifungal effect against *Cryptococcus neoformans* and *Trichophytum rubrum* with MIC values of 72 µg/mL (Table 2). For the remaining dermatophytes, the oil was also effective with MIC values ranging from 144 µg/mL to 288 µg/mL. In addition, relevant antifungal effects were observed for *Aspergillus* and the majority of *Candida* strains (Table 2). Interestingly for the majority of the tested stains, the oil showed similar MIC and MLC values, thus presenting a fungicidal effect (Table 2). This is quite relevant since azoles, namely flucanozole, an antifungal drug widely used in the clinic, shows primarily fungistatic effects. Indeed, this antifungal drug, for the majority of the tested strains, is able to inhibit fungal growth but at the MIC it is not lethal to the fungi.

#### 2.2.2. The Essential Oil was Effective against *Candida albicans* Virulent Factors

*Candida albicans* is able to switch from yeast-to-hypha morphology promoting tissue invasiveness [33] and to form biofilms, an organized structure more resistant to antifungal agents than planktonic fungi [34]. These features play a pivotal role in *C. albicans* pathogenesis and virulence, thus constituting promising therapeutic targets. Therefore, the present study aimed to assess the effect of the essential oil on both features (Figure 1 and Figure 2). As shown in Figure 1, the essential oil strongly inhibited germ tube formation in *C. albicans* at concentrations well below its respective MIC. Interestingly, at concentrations as low as MIC/8 (72 μg/mL) the oil inhibited germ tube formation by ca. 40% and at MIC/4 (144 μg/mL), 88% of inhibition was observed (Figure 1a). Fluconazole failed to accomplish this up to 200 μg/mL, as pointed out in Figure 1b, thus highlighting the relevance of our results. 

The capacity of the essential oil to decrease biofilm mass and biofilm viabilitywas also tested as well as that of fluconazole (Figure 2). Overall, the oil was effective and decreased biofilm viability at concentrations similar to the respective MIC and at a slightly higher concentration, the oil also disrupted mature biofilms (Figure 2a). Fluconazole, on the other hand, was only effective in decreasing *C. albicans* viability at a much higher concentration than its respective MIC and was ineffective in reducing biofilm mass even at concentration 200× higher than its MIC, as shown in Figure 2b.

#### 2.2.3. The Essential Oil Modified the Ultrastructure of *Candida albicans* and *Trichophyton rubrum*

Bearing in mind the promising fungicidal effect observed against *T. rubrum* as well as the inhibition of virulence factors registered in *C. albicans*, both strains were selected to further elucidate the effect of the essential oil at an ultrastructural level. For this, transmission electron microscopy analysis were performed. Figure 3 and Figure 4 compile representative images of the alterations observed in *C. albicans* and *T. rubrum*, respectively, in comparison to non-treated fungal cells. Overall, untreated cells showed a well-defined surface with and integral cell wall (Figure 3a,b; Figure 4a,b) while treated cells (Figure 3c–h; Figure 4c–h) presented a structural disorganization, particularly at the cell wall (e.g., Figure 3c and Figure 4c), for *C. albicans* and *T. rubrum* respectively. These alterations can be due to an impairment in the deposition of cell wall constituents. In *C. albicans* cytoplasmic disintegration was also observed (e.g., Figure 3h). Particularly for *T. rubrum* an increased number of vacuoles was observed with the accumulation of electron dense material (Figure 4c–h), reflecting a defensive mechanism of the fungus. As the number of vacuoles significantly increased with the essential oil treatment, it seems that autophagy is activated [35]. Indeed, autophagic structures were observed (Figure 4h), thus suggesting that the oil activates this pathway as a protective mechanism of the fungi.

### 2.3. Anti-Inflammatory Activity

Chronic inflammation contributes to the successful colonization of the host tissues by pathogenic fungi, thus compromising disease suppression. Indeed, an antifungal drug with inherent anti-inflammatory activity, would enable a rapid symptomatic relief while providing an effective treatment. Therefore, we also assessed the anti-inflammatory potential of the essential oil using an in vitro model of lipopolysaccharide (LPS)-stimulated macrophages. The effect on NO production, a canonical marker of inflammation, was analyzed by measuring the accumulation of nitrites in the culture medium. Moreover, the effect of the oil on macrophages’ viability was also tested (Figure 5). Expectably, our results show that macrophages without LPS stimuli produced residual levels of nitrites (<1 µM) and this value increased to 35.25 ± 3.91 µM in the presence of LPS (Figure 5a). Interestingly, nitrite production was significantly reduced in the presence of the essential oil, attaining a decrease of more than 51% with 576 µg/mL of the oil (17.15 ± 3.35 µM) and a decrease of 33% with half the concentration (23.53 ± 3.93 µM; Figure 5a). Importantly, all the tested concentrations of the essential oil did not affect macrophages’ viability (Figure 5b), thus confirming a safety profile of the oil at concentrations exhibiting pharmacological activity.

## 3. Discussion

*Bupleurum rubrum* subp. *paniculatum* essential oils from Portugal showed high amounts of monoterpene hydrocarbons with α-pinene, β-pinene and limonene standing out as the main compounds. Similar compounds were also reported for essential oils obtained from plants collected in Montes Toledo, Spain, although in these plants myrcene was also found in high amounts (26.2%) [36]. These differences might be due to different growing locations with diverse abiotic factors that may influence the biosynthesis of essential oils. In a previous work, we developed a protocol for the in vitro propagation of *Bupleurum* species and observed the presence of the major compounds inside secretory ducts of the in vitro propagated plantlets of *B. rigidum* subsp. *paniculatum* [37]. Since secondary metabolites signature in spontaneous plants was also found in the in vitro propagated plants, this technique could enable a large-scale production of this species, thus reducing the influence of abiotic factors and enabling the production of uniform essential oils. Furthermore, this approach could allow a year-round plant supply.

To the best knowledge of the authors, no reports on the bioactive potential of *B. rigidum* subsp. *paniculatum* have been conducted. Nevertheless, the antimicrobial activity of other *Bupleurum* species has already been reported. For example, the essential oil isolated from *B. marginatum*, rich in tridecane, undecane, pentadecane, β-caryophyllene and β-caryophyllene oxide, showed a MIC of 4 mg/mL towards two strains of *C. albicans* and inhibited several bacterial strains, including methicillin-resistant *S. aureus* (MIC = 0.063–4 mg/mL) [22]. The antifungal activity of *B. gibraltaricum*, characterized by sabinene, α-pinene and 2,3,4-trimethylbenzaldehyde, was demonstrated for *Plasmopara halstedii* through fungal sporulation inhibition at 5 mL/L [30]. The essential oil from *B. longiradiatum*, having thymol, butylidene phthalide, 5-indolol, heptanal, 4-hydroxy-2-methylacetophenone, 4,5-diethyl-octane, borneol and hexanoic acid as main compounds, was able to inhibit the growth of several bacteria with MIC ranging from 250–500 μg/mL [27]. The essential oil herein presented showed a broad antifungal activity, very effective against *Cryptococcus neoformans* and *Trichophyton rubrum* (MIC = 72 µg/mL). The antifungal effect of the essential oil might be attributed to the presence of several compounds with strong antifungal activity, namely α-pinene, limonene, β-pinene and myrcene [38,39,40,41,42,43,44,45,46]. Regarding the anti-virulent effects of *Bupleurum* spp., no studies have been conducted until now. Nevertheless, the major compounds of the essential oil were able to decrease the yeast-to-hypha transition in *C. albicans*. Indeed, β-pinene [47,48] and limonene [48] were described as germ tube inhibitors. Similarly, the capacity of the oil to disrupt preformed *C. albicans* biofilms might also be attributed to the presence of limonene and β-pinene as both compounds have been described as effective in disrupting mature biofilms. Nevertheless, the influence of minor compounds cannot be discarded. Indeed, myrcene and *p*-cymene were also able to inhibit the formation of *C. albicans* biofilms [48].

Regarding ultrastructural characterizations, studies on dermatophytes are scarce, thus highlighting the relevance of our results. In *Candida* species, essential oils tend to act on ergosterol synthesis, modify fungal cell morphology, modulate membrane permeability and produce reactive oxygen species [49]. In accordance, our observations showed alterations in the cell wall structure as well as cytoplasmic disorganization, thus contributing to an overall modification of fungal cell morphology. Similar effects were reported for conventional antifungal drugs, namely caspofungin alone or in conjugation with pyroquilon in *Alternaria infectoria* [50], suggesting a similar mechanism of action for *B. rigidum* subsp. *paniculatum* essential oils. Regarding *T. rubrum*, our results clearly show that the essential oil induces an increase in the number of vacuoles, thus suggesting an activation of autophagic pathways. Similar results were shown for *Microsporum gypseum* treated with resorcinol derivatives [35].

Furthermore, the present study reports for the first time the anti-inflammatory potential of *B. rigidum* subsp. *paniculatum*. Nevertheless, other studies have pointed out promising effects for other *Bupleurum* species. For example, the essential oil from *B. fruticosum*, rich in α- and β-pinene, thymol and carvacrol, was able to decrease the edema in a carrageenan-induced edema model [23] and that of *B. frutiscens*, characterized by α-pinene and β-caryophyllene, also showed effects in paw edema induced by carrageenan and prostaglandin (PG) E1 [24]. Moreover, *B. gibraltaricum* essential oil with high amounts of Δ-3-carene and α-pinene, showed anti-inflammatory effects in both carrageenan-induced paw edema and granuloma-induced inflammation [51,52]. Additionally, the production of PGE2 and the activity of lipoxygenase was decreased in the presence of *B. marginatum* essential oil [22]. Overall, these results corroborate the anti-inflammatory effects observed in the present study as all species present similar major compounds to those found in *B. rigidum* subsp. *paniculatum*, namely pinene isomers. Accordingly, several reports have shown the anti-inflammatory effect of isolated major compounds including α- and β-pinene as well as limonene [53,54,55]. In addition, minor compounds, such as 1,8-cineole [56], camphor [56,57] and γ-terpinene [58] have also been assessed in a plethora of assays, thus suggesting that these compounds may contribute to *B. rigidum* subsp. *paniculatum* anti-inflammatory potential.

## 4. Conclusions

Overall, the present study provides evidence, for the first time, of the antifungal and anti-inflammatory potential of *B. rigidum* subsp. *paniculatum* essential oil from Portugal. The oil showed high amounts of monoterpene hydrocarbons (94.4–96.6%), namely α-pinene (29.0–36.0%), β-pinene (26.1–30.7%) and limonene (10.5–13.5%) and demonstrated a broad fungicidal effect, particularly against *Cryptococcus neoformans* and *Trichophyton rubrum*. Moreover, a significant germ tube inhibition as well as a disruption of mature biofilms for *C. albicans* was pointed out and for both *C. albicans* and *T. rubrum* significant ultrastructural alterations were observed. Finally, *B. rigidum* subsp. *paniculatum* essential oil also evidenced anti-inflammatory activity, without affecting macrophages´ viability, further endorsing its exploitation as a natural source of bioactive compounds for the development of drugs with concomitant antifungal and anti-inflammatory effects. Nevertheless, further in vivo studies are needed to validate the efficacy of the essential oil. Moreover, the clinical relevance of our results depends on the availability of the essential oil on target organs. Therefore, pharmacokinetic studies are essential to link in vitro effects to in vivo efficacy.

## 5. Materials and Methods

### 5.1. Plant Material

Aerial parts of two samples of *B. rigidum* subsp. *paniculatum* (~1 kg) were collected from field-growing plants in the flowering stage in central Portugal [A—Coimbra region (40°14′52.4″ N 8°25′14.1″ W), B—Fatima region (39°37′28.6″ N 8°41′19.7″ W)]. Voucher specimens were deposited at the herbarium of the Department of Life Sciences of the University of Coimbra (COI).

### 5.2. Essential Oil Isolation and Chemical Analysis

The essential oils were obtained by hydrodistillation (3 h) using a Clevenger-type apparatus as described in the European Pharmacopoeia [59]. Their chemical characterization was carried out using both gas chromatography (GC) and gas chromatography coupled to mass spectrometry (GC-MS).

A Hewlett-Packard 6890 (Agilent Technologies, Palo Alto, CA, USA) chromatograph with a single injector, two flame ionization detectors and a HP GC ChemStation Rev. A.05.04 data handling system was used for GC analysis. Two columns: polydimethylsiloxane (SPB-1) and polyethylene glycol (SupelcoWax-10) with 30 m × 0.20 mm and a film thickness 0.20 µm were used and GC parameters were as follows: oven temperature: 70–220 °C (3 °C/min), 220 °C (15 min); injector temperature: 250 °C; carrier gas: helium, with a linear velocity of 30 cm/s; splitting ratio 1:40; detectors temperature: 250 °C. A Hewlett-Packard 6890 gas chromatograph (SPB-1 column), interfaced with an Hewlett-Packard mass selective detector 5973 (Agilent Technologies, Centerville Road Wilmington, DE, USA) with a HP Enhanced ChemStation software, version A.03.00 (Agilent Technologies, Palo Alto, CA, USA) was used for GC–MS analysis. MS parameters: source temperature: 230 °C; MS quadruple temperature: 150 °C; ionization energy: 70 eV; ionization current: 60 µA; scan range: 35–350 units; scans per second: 4.51; interface temperature: 250 °C.

Retention indices and mass spectra were used to identify the compounds with the first being compared to samples from the Laboratory of Pharmacognosy database of the Faculty of Pharmacy, University of Coimbra and the later with that of reference spectra from a home-made library or from literature data. GC peak area was used to calculate relative amounts of the identified compounds.

### 5.3. Antifungal Activity

#### 5.3.1. Fungal Strains

Collection and clinical strains were used to assess the antifungal effect of the essential oil, including yeasts: *Candida albicans* ATCC 10231, *C. guillermondii* MAT23, *C. krusei* H9, *C. parapsilosis* ATCC 90018, *C. tropicalis* ATCC 13803, and *Cryptococcus neoformans* CECT 1078; dermatophytes: *Epidermophyton floccosum* FF9, *Microsporum canis* FF1, *M. gypseum* CECT 2908, *Trichophyton mentagrophytes* FF7, *T. mentagrophytes* var. *interdigitale* CECT 2958, *T. rubrum* CECT 2794, and *T. verrucosum* CECT 2992 and *Aspergillus* strains including *A. flavus* F44, *A. fumigatus* ATCC 46645 and *A. niger* ATCC 16404.

The fungal isolates were identified by standard microbiological methods and stored at −80 °C on Sabouraud broth with 20% glycerol, being inoculated on Sabouraud dextrose agar (SDA) to ensure optimal growth and purity, prior to each assay.

#### 5.3.2. Fungal Growth

To assess the effect of the essential oil on fungal growth, two parameters were assessed: the minimal inhibitory concentration (MIC) and the minimal lethal concentration (MLC). A broth macrodilution method was used according to the Clinical and Laboratory Standards Institute (CLSI) reference protocols M27-A3 and M38-A2, for yeasts and filamentous fungi, respectively. Serial dilutions of the oil (up to 0.036 µg/mL) were tested as well as a negative control (medium without fungi) and a positive control (inoculated medium with DMSO 1%).

#### 5.3.3. Fungal Ultrastructure

*C. albicans* and *T. rubrum* were selected to perform the transmission electron microscopy studies. Cultures untreated (DMSO 1%) or treated with the essential oil (MIC) were used. Fungi cultures were collected and centrifuged at 3000 rpm for 5 min. Pellet cells were fixed using 2.5% glutaraldehyde in 0.1 M sodium cacodylate buffer (pH 7.2) for 2 h and post-fixed in 1% osmium tetroxide during 1 h. Following rinsing, samples were embedded in 2% molten agar, dehydrated in ethanol (30–100%), and impregnated and included in epoxy resin (Sigma Aldrich, St. Louis, MO, USA). Ultrathin sections (70 nm) were obtained and stained with uranyl acetate 2%, for 5 min followed by lead citrate 0.2% for 5 min. Observations were carried out on a FEI Tecnai G2 Spirit Bio Twin at 100 kV.

#### 5.3.4. *Candida albicans* Germ Tube Formation

To assess the effect of the oil on the yeast–hyphae transition, cell suspensions of *C. albicans* ATCC 10231, from overnight cultures on SDA, were prepared in NYP medium [N-acetylglucosamine (10–3 mol/L), Yeast Nitrogen Base (3.35 g/L), proline (10–3 mol/L), NaCl (4.5 g/L), and pH 6.7 ± 0.1] [60]. The suspensions were adjusted to a density of 1.0 ± 0.2 × 10^6^ CFU/mL and then distributed into glass test tubes (990 μL). Then, subinhibitory concentrations of the essential oil or fluconazole (10 µL) were added and incubated without agitation for 3 h, at 37 °C. The number of germ tubes (germinating protuberance at least as long as the diameter of the blastopore) were recorded using a light microscope (40×) and DMSO 1% (*v*/*v*) was used as a positive control.

#### 5.3.5. *Candida albicans* Biofilm Disruption

Serial dilutions of the essential oil (1125–72 µg/mL) and fluconazole (200–1 µg/mL) were used to assess their effect on mature biofilms, as reported elsewhere [61]. *C. albicans* density was adjusted to 1 × 10^6^ CFU/mL and 100 μL of this suspension added to 96-well microtiter plates and incubated for 24 h at 37 °C for biofilm adhesion and formation, followed by 24 h at 37 °C in the presence or absence of the essential oils.

Biofilm viability was determined using the XTT/menadione metabolic assay. Briefly, menadione (10 mM in acetone) was added to a XTT solution (1 mg/mL in PBS) to a final concentration of 4 µM. A volume of 100 µL of this solution was added to each well and incubated for 2 h at 37 °C in the dark. The absorbance was measured at 490 nm and cell viability expressed as [(AbsT/AbsC) × 100], where AbsT represents the absorbance of the treated cells and AbsC the absorbance of control cells (without essential oil or fluconazole).

Biofilm biomass was assessed using the crystal violet assay. Following the treatment period, performed as referred above, the culture medium was removed and cells were fixed with methanol 99% for 15 min. The supernatant was discarded, and the wells air dried. Then, 100 µL of crystal violet (CV) solution (0.02%) was added to each well and left to stain the biofilm for 15 min. After CV removal, the wells were washed twice with sterile water to remove excessive reagent. Then, 150 µL of acetic acid 33% were added to release the stain from the cells and the supernatant was transferred to a 96-well microtiter plate. The absorbance was then read at 620 nm using a microtiter plate reader. Biomass decrease was determined as [(AbsT/AbsC) × 100], where AbsT represents the absorbance of each treated well and AbsC the absorbance in the control wells. Negative and positive controls comprising biofilm-free and oil-free wells, respectively, were also included.

### 5.4. Anti-Inflammatory Activity

#### 5.4.1. Cell Culture

The mouse macrophage cell line, Raw 264.7 (ATCC number: TIB-71) was kindly supplied by Dr. Otilia Vieira (CEDOC, Faculdade de Ciências Médicas, Universidade Nova de Lisboa). The cells were cultured on endotoxin-free Dulbecco’s Modified Eagle Medium (DMEM) supplemented with 10% (*v*/*v*) non-inactivated fetal bovine serum, 3.02 g/L sodium bicarbonate, 100 µg/mL streptomycin and 100 U/mL penicillin at 37 °C in a humidified atmosphere with 5% CO_2_. Morphological appearance of macrophages was microscopically monitored during the assays.

#### 5.4.2. Assessment of Nitric Oxide (NO) Production

To evaluate the anti-inflammatory potential of the oils, the in vitro model of lipopolysaccharide (LPS)-stimulated macrophages was used. The cells (0.6 × 10^6^ cells/well) were cultured in a 48-well microplate and left to stabilize for 12 h. The cells were then incubated for 24 h in the culture medium alone (control) or with different concentrations of the essential oils (144–576 µg/mL).The production of nitric oxide (NO) was measured by the accumulation of nitrites in the culture medium, using the colorimetric Griess reaction. Briefly, 170 µL of culture supernatants were added to an equal volume of Griess reagent [0.1% (*w*/*v*) N-(1-naphthyl)-ethylenediamine dihydrochloride and 1% (*w*/*v*) sulphanilamide containing 5% (*w*/*v*) H_3_PO_4_] and maintained for 30 min, in the dark. Absorbance was measured using an ELISA automatic microplate reader (SLT, Austria) at 550 nm and the nitrite concentration determined from a regression analysis prepared with serial dilutions of sodium nitrite.

#### 5.4.3. Assessment of Cell Viability

Cell respiration, an indicator of cell viability, was performed using a colorimetric assay with 3-(4,5-dimethylthiazol-2-yl)-2,5-diphenyl tetrazolium bromide (MTT). The cells were cultured and treated as referred above. After cells treatments, 43 µL of MTT solution (5 mg/mL in phosphate buffered saline) were added to each well and the microplates were further incubated at 37 °C for 15 min, in a humidified atmosphere with 5% CO_2_. Supernatants were centrifuged (1000× *g* during 5 min) to recover viable cells. To dissolve formazan crystals formed in adherent cells in the microplates, 300 µL of acidified isopropanol (0.04 M HCl in isopropanol) were added to each cell and recovered to the respective tube containing the pellet formed after centrifugation. Quantification of formazan was performed using an ELISA automatic microplate reader (SLT, Salzburg, Austria) at 570 nm.

### 5.5. Statistical Analysis

The results presented as mean ± SD are from experiments performed at least in triplicate. The statistical test one-way ANOVA followed by Dunnett’s Multiple Comparison Test were performed using GraphPad Prism, version 6 (GraphPad Software, San Diego, CA, USA).

## Figures and Tables

**Figure 1 antibiotics-10-00592-f001:**
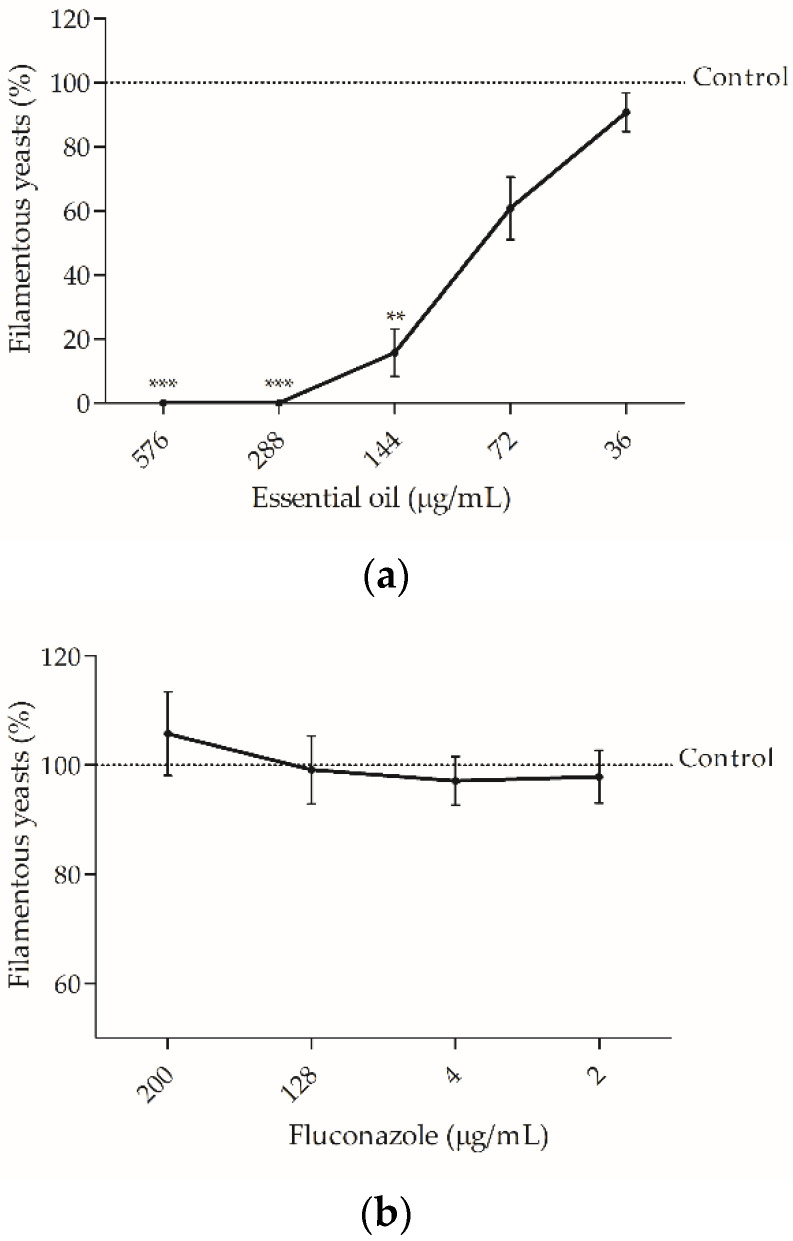
Filamentation percentage of *Candida albicans* ATCC 10231 treated with different concentrations of *Bupleurum rigidum* subp. *paniculatum* essential oil (**a**) or treated with fluconazole (**b**). One-way ANOVA followed by Dunnett’s Multiple Comparison Test, ** *p* < 0.01; *** *p* < 0.001, compared to control.

**Figure 2 antibiotics-10-00592-f002:**
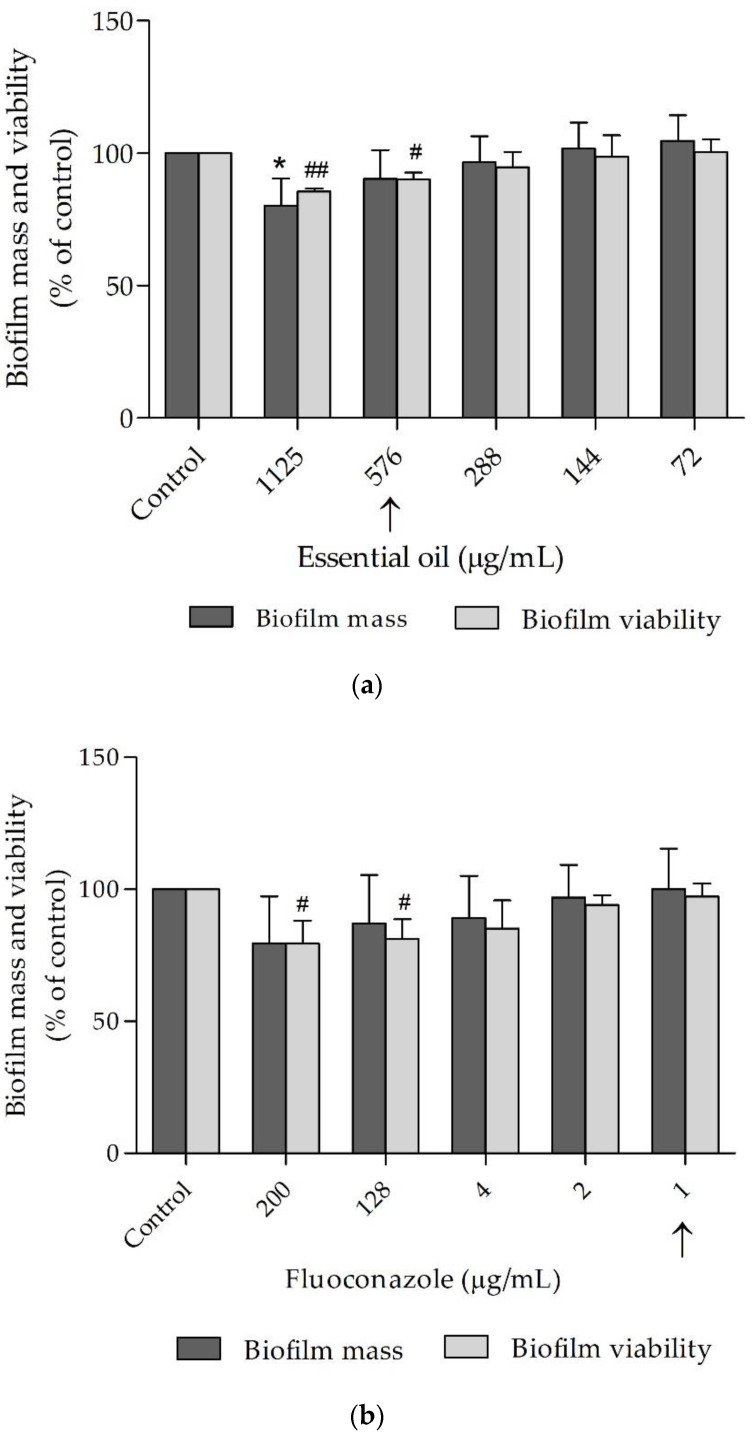
Effect of *Bupleurum rigidum* subp. *paniculatum* essential oil (**a**) and fluoconazole (**b**) on *Candida albicans* ATCC 10231 biofilm biomass and viability. One-way ANOVA followed by Dunnett’s multiple comparison test (performed in separate on each feature), * *p* < 0.05, compared to biofilm mass control; # *p* < 0.05 and ## *p* < 0.01, compared to biofilm viability control; arrows indicate MIC value.

**Figure 3 antibiotics-10-00592-f003:**
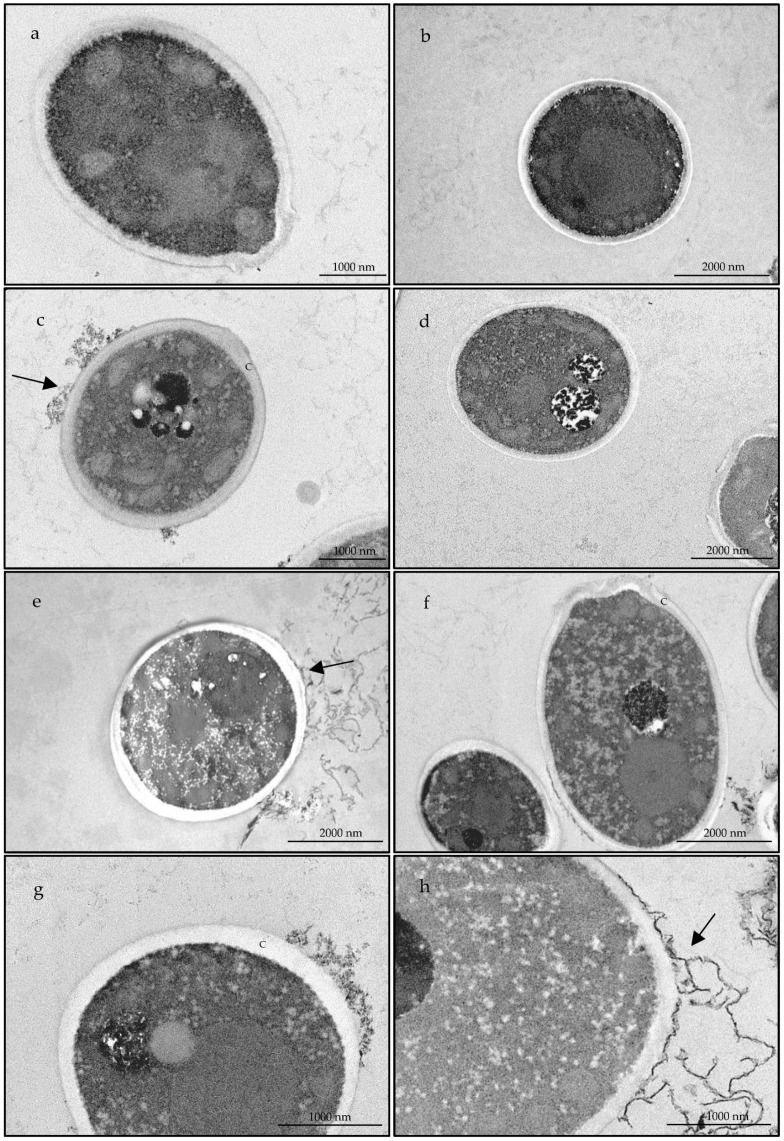
Transmission electron microscopic observations of *Candida albicans* in the presence of DMSO (**a**,**b**) or treated with of 72 µg/mL of *Bupleurum rigidum* subsp. *paniculatum* essential oil (**c**–**h**). Arrow points to fragments from the cell wall; c—cell wall.

**Figure 4 antibiotics-10-00592-f004:**
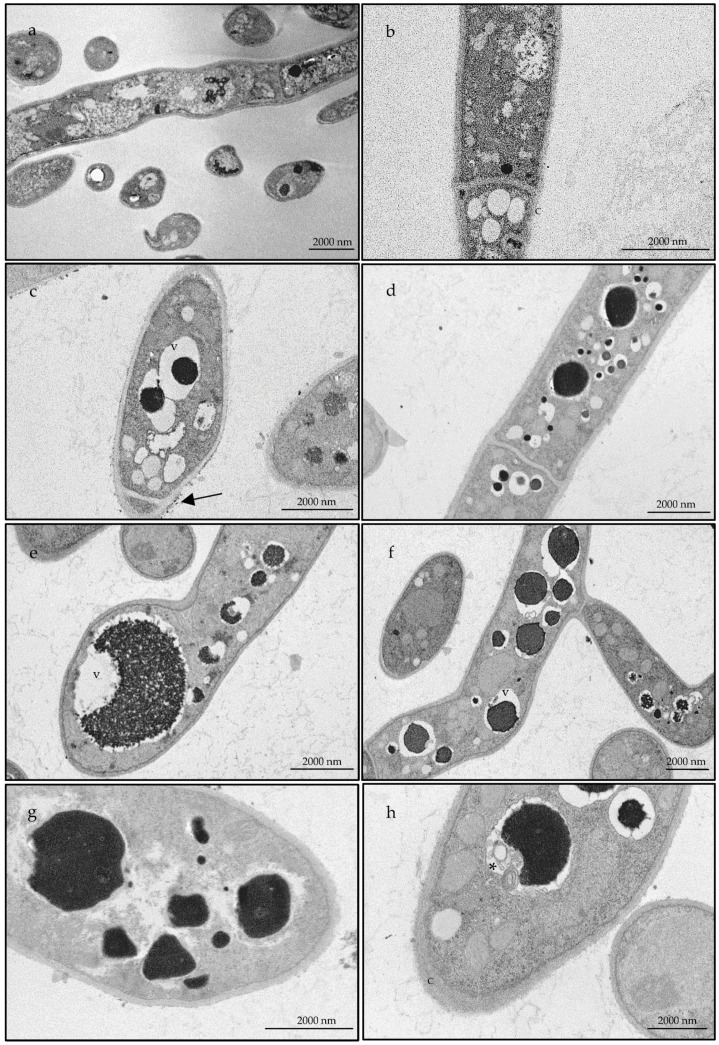
Transmission electron microscopic observations of *Trichophyton rubrum* in the presence of DMSO (**a**,**b**) or treated with of 576 µg/mL of *Bupleurum rigidum* subsp. *paniculatum* essential oil (**c**–**h**). Arrow points to fragments from the cell wall; * shows an autophagic structure; c—cell wall, v—vacuoles.

**Figure 5 antibiotics-10-00592-f005:**
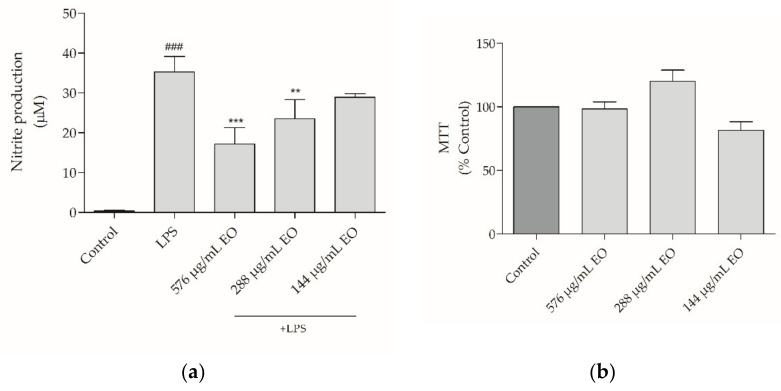
Effect of *Bupleurum rigidum* subp. *paniculatum* essential oil on NO production (**a**) and cell viability (**b**). One-way ANOVA followed by Dunnett’s Multiple Comparison Test, ### *p* < 0.001, compared to control; ** *p* < 0.01; *** *p* < 0.001, compared to LPS. EO—essential oil.

**Table 1 antibiotics-10-00592-t001:** Chemical composition of Portuguese *Bupleurum rigidum* subsp. *paniculatum* essential oil.

Compound *	RI SPB-1 ^a^	RI SW 10 ^b^	Coimbra (%)	Fátima (%)
α-Thujene	922	1029	0.3	0.1
α-Pinene	930	1030	36.0	29.0
Camphene	943	1077	0.2	0.4
Fenchene	943	1067	0.1	0.1
Sabinene	964	1128	2.0	2.6
β-Pinene	970	1118	26.1	30.7
Myrcene	981	1160	1.8	10.0
α-Phellandrene	997	1171	0.3	0.1
α-Terpinene	1006	1183	t	t
*p*-Cymene	1009	1271	0.2	t
Limonene	1021	1204	10.5	13.5
β-Phellandrene	1021	1214	9.4	4.5
Z-β-Ocimene	1025	1235	4.1	1.0
*E*-β-Ocimene	1035	1253	5.1	2.0
γ-Terpinene	1046	1249	0.3	0.2
Terpinolene	1076	1288	0.1	0.1
**Monoterpene hydrocarbons**			96.6	94.4
Linalool	1082	1543	t	t
α-Pinene-xide	1077	1370	t	t
*allo-*Ocimene	1118	1370	0.1	0.1
Terpinene-4-ol	1159	1595	0.1	t
Myrtenal	1176	1786	0.1	t
α-Terpineol	1168	1692	t	t
Bornyl acetate	1266	1578	0.1	0.1
**Oxygen containing monoterpenes**			0.5	0.3
α-Copaene	1364	1487	0.1	t
β-Cubebene	1380	1538	t	t
*E*-Caryophyllene	1408	1590	0.2	0.1
γ-Elemene	1417	2136	t	t
α-Humulene	1442	1665	t	t
Germacrene-D	1466	1699	1.7	0.5
Bicyclogermacrene	1478	1727	0.2	0.1
δ-Cadinene	1508	1751	0,1	t
**Sesquiterpene hydrocarbons**			2.4	0.8
Spathulenol	1551	2112	t	t
Caryophyllene oxide	1557	1968	t	t
**Oxygen containing sesquiterpenes**			0.1	0.1
**Total identified**			99.6	95.6

* Compounds listed in order of elution in the SPB-1 column; t < 0.05%. ^a^ RI SPB 1: GC retention indices relative to C9-C23 n-alkanes on the SPB-1 column. ^b^ RI SW 10: GC retention indices relative to C9-C23 n-alkanes on the Supelcowax-10 column.

**Table 2 antibiotics-10-00592-t002:** Antifungal activity of *Bupleurum rigidum* subsp. *paniculatum* essential oil.

Strains	Essential Oil		Fluconazole	
	MIC	MLC	MIC	MLC
*Candida albicans* ATCC 10231	576	576	1	>128
*Candida tropicalis* ATCC 13803	1125	1125	4	>128
*Candida krusei* H9	288	288	64	64–128
*Candida guillermondii* MAT23	288	288	8	8
*Candida parapsilosis* ATCC 90018	576	576	1	1–2
*Cryptococcus neoformans* CECT 1078	72	144	16	128
*Trichophytum mentagrophytes* FF7	288	288	16–32	32–64
*Trichophytum mentagrophytes* var. *interdigitale* CECT 2958	288	288	128	≥128
*Trichophyton rubrum* CECT 2794	72	72	16	64
*Trichophytum verrucosum* CECT 2992	288	288	128	>128
*Microsporum canis* FF1	144	144	128	128
*Microsporum gypseum* CECT 2905	288	288	128	>128
*Epidermophyton floccosum* FF9	144	144	16	16
*Aspergillus niger* ATCC 16404	288	1125	n.t	n.t
*Aspergillus fumigatus* ATCC 46645	576	576	n.t	n.t
*Aspergillus flavus* F44	576	576	n.t	n.t

MIC—minimal inhibitory concentration and MLC—minimal lethal concentration determined by a macrodilution method and expressed as µg/mL; n.t—not tested.

## Data Availability

Data is contained within the article.

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
