# Peer review of "Antifungal and Anti-Inflammatory Potential of Bupleurum rigidum subsp. paniculatum (Brot.) H.Wolff Essential Oil"

_antibiotics, 2021, doi:10.3390/antibiotics10050592_

Round 1

Reviewer 1 Report

The manuscript "Bioactive potential of Bupleurum rigidum subsp. paniculatum
essential oil" aims to assess, for the first time, the antifungal and anti-inflammatory potential of Bupleurum subsp. paniculatum essential oil from Portugal. 

The manuscript is interesting. However, several issues need attention before publication. 

Title: The title is too broad. Biological potential means several things. It is better to narrow to the two properties assessed. Also, check the proper use of the italic.

Abstract: Please, refer to specific amounts (Lines 22-23) to the word "high."The expression could be described more specifically. 

Line 30: What is LPS?

Line 32-34: Describe the lines similar to the conclusion section (lines 447-451).

Introduction: It is necessary to decrease the number of references and to update some of them (there are several ancient references). 

Lines 108-113 are repetitive. Please check.

Results

Table 1. I suggest adding a column to identify monoterpenes, sesquiterpenes, oxygenated monoterpenes, and oxygenated sesquiterpenes. I believe grouping the compounds in this way is better for the reader than for elution order. 

Line 121: Please include the amount instead of using "high amounts." 

Line 246: Please add a reference to support the statement.

Lines 142-145: An in-depth explanation is needed in the discussion section.

Line 167: Thus, instead of This.

Figure 5b. What does it mean an MTT (% control) superior to 100% for 0.32 uL/mL of EO?

Conclusions: It is better to add the specific amounts instead of expressions like "significant amounts" or "high amounts 

The section of conclusions should be number 4. Materials and methods should be 5.."

Material and methods: Please double check units for all the measurements, according to the international units system 

Section 4.4: What does it mean +- SEM?

Author Response

Reviewer #1

The manuscript "Bioactive potential of Bupleurum rigidum subsp. paniculatum
essential oil" aims to assess, for the first time, the antifungal and anti-inflammatory potential of Bupleurum subsp. paniculatum essential oil from Portugal. The manuscript is interesting. However, several issues need attention before publication. 

  1. Title: The title is too broad. Biological potential means several things. It is better to narrow to the two properties assessed. Also, check the proper use of the italic.

The title has been modified to Antifungal and anti-inflammatory potential of Bupleurum rigidum subsp. paniculatum (Brot.) H.Wolff essential oil. The species name was written in italic.

  1. Abstract: Please, refer to specific amounts (Lines 22-23) to the word "high." The expression could be described more specifically. 

To describe more specifically the amounts of monoterpenes, the percentage range for each of the major compounds was included.

  1. Line 30: What is LPS?

LPS refers to lipopolysaccharide, an endotoxin found in the outer membrane of Gram-negative bacteria. The abbreviation LPS was replaced by the full word to avoid misunderstandings.

  1. Line 32-34: Describe the lines similar to the conclusion section (lines 447-451).

The lines in the abstract were rephrased to ‘Overall, the present study confirmed the potential of Bupleurum rigidum subsp. paniculatum essential oil, further endorsing its exploitation as a natural source of bioactive compounds for the development of drugs with both anti-fungal and anti-inflammatory effects.’

  1. Introduction: It is necessary to decrease the number of references and to update some of them (there are several ancient references).

The number of references were decreased and several updated.

  1. Lines 108-113 are repetitive. Please check.

The sentences were rephrased to avoid repetitions. We intend to indicate the aim of the study and also point out the main results as required in the instructions for publication.

Results

  1. Table 1. I suggest adding a column to identify monoterpenes, sesquiterpenes, oxygenated monoterpenes, and oxygenated sesquiterpenes. I believe grouping the compounds in this way is better for the reader than for elution order. 

The compounds were grouped as suggested and the amounts of each group added, providing an easier way for the reader to identify the class of compounds.

  1. Line 121: Please include the amount instead of using "high amounts." 

The amounts of the monoterpene hydrocarbons as well as those of the main compounds were included.

  1. Line 246: Please add a reference to support the statement.

A reference was added.

  1. Lines 142-145: An in-depth explanation is needed in the discussion section.

The sentence: ‘Interestingly for the majority of the tested stains, the oil showed a fungicidal effect (Table 2). This is quite relevant since azoles, namely flucanozole, an antifungal drug widely used in the clinic, shows primarily fungistatic effects’

Was changed to:

Interestingly for the majority of the tested stains, the oil showed similar MIC and MLC values, thus presenting a fungicidal effect (Table 2). This is quite relevant since azoles, namely flucanozole, an antifungal drug widely used in the clinic, shows primarily fungistatic effects. Indeed, this antifungal drug for the majority of the tested strains is able to inhibit fungal growth but at the MIC is not lethal to the fungi.

  1. Line 167: Thus, instead of This.

The mistake was corrected.

  1. Figure 5b. What does it mean an MTT (% control) superior to 100% for 0.32 uL/mL of EO?

The EO at 0.32 uL/mL may induce cell proliferation that would explain a viability higher than 100%.

  1. Conclusions: It is better to add the specific amounts instead of expressions like "significant amounts" or "high amounts’

The specific amounts of the compounds were included.

  1. The section of conclusions should be number 4. Materials and methods should be 5."

The mistake was corrected.

  1. Material and methods: Please double check units for all the measurements, according to the international units system 

The units were all double-checked and corrected according to the international units system.

  1. Section 4.4: What does it mean +- SEM?

Mean and standard error of mean (SEM) are used to describe the variability within the sample.

Reviewer 2 Report

The study is quite impressive and scientifically elaborated. I advise some corrections:

- I suggest for authors to verify the parameters and conditions of the analysis and the machine information (name, manufacturers ...).

- Add the abundance % of the major constituent in the abstract section. The same remark in the line 121.

- Line 122: justify why you choice sample from Coimbra because the abundance of compounds in the two samples was different.

- Line 139: why you represent the MIC values with µL/mL not with µg or mg/mL. I suggest changing it.

- The plant material section, please the geographical localization GPS of the site where the samples were collected. Time of extraction and the quantity of plant used in the extraction.

- Improve the quality of graphics.

- Ensure the uniformity in the units (e.g. mg/mL, µL) throughout the MS.

- Correct some English mistakes before publication.

- Check the references in accordance with the journal style.

Author Response

Reviewer #2

The study is quite impressive and scientifically elaborated. I advise some corrections:

  1. I suggest for authors to verify the parameters and conditions of the analysis and the machine information (name, manufacturers ...).

The parameters and conditions of oil analysis were checked and more information provided.

  1. Add the abundance % of the major constituent in the abstract section. The same remark in the line 121.

The abundance of the major compounds of the essential oils were added in the abstract section and also in line 121.

  1. Line 122: justify why you choice sample from Coimbra because the abundance of compounds in the two samples was different.

Coimbra region presents the largest population of Bupleurum rigidum subsp. paniculatum and therefore was selected to perform the bioactive studies. A better explanation is provided.

The sentence ‘Considering the similar composition between both samples, the one from Coimbra (largest population) was selected to assess the antifungal and anti-inflammatory potential of the essential oil.’

Was changed to:

‘The essential oil from Coimbra region represents a much larger population and therefore was selected to assess the antifungal and anti-inflammatory potential of the oil.’

  1. Line 139: why you represent the MIC values with µL/mL not with µg or mg/mL. I suggest changing it.

MIC values were altered to µg/mL as this was also suggested by reviewer #3.

  1. The plant material section, please the geographical localization GPS of the site where the samples were collected. Time of extraction and the quantity of plant used in the extraction.

The GPS localization of the site where plants were collected is provided. The extraction time was already provided in the subsection Essential oil isolation and chemical analysis and the quantity of plant used was added (1kg).

  1. Improve the quality of graphics.

The quality of the graphs was improved.

  1. Ensure the uniformity in the units (e.g. mg/mL, µL) throughout the MS.

The units were corrected according to the SI system.

  1. Correct some English mistakes before publication.

English errors were corrected

  1. Check the references in accordance with the journal style.

The references were formatted according to the journal style.  

Reviewer 3 Report

The paper is interesting and worth publishing, but a number of corrections have to be made, as discussed below.

Line 2: Please use the complete name in the title, “Bupleurum rigidum subsp. paniculatum (Brot.) H.Wolff”. This should also be used for the first time in the full text and abstract.

Line 21: please correct the name to its complete form, as above.

Line 29: it is probably the effect “of the oil”, not “on the oil”.

Line 32: please correct the name to “Bupleurum rigidum subsp. paniculatum”.

Lines 59-60: “undesirable side effects” are actually “safety issues”, therefore one of the two is sufficient here.

Line 113: the meaning of “antivirulent” is confusing, therefore we suggest replacing it with a clearer term. The same holds true for other occurrences of the term.

Lines 508: the DMSO concentration should be stated.

Lines 590: for good reasons discussed in (https://journals.sagepub.com/doi/full/10.1177/0253717620933419) I would recommend replacing SEM with standard deviation.

Line 118: the 0.6% yield was identical for the two samples? If so, this should be stated clearly, so as to avoid confusions.

Lines 139-141: the fact that MICs are expressed as ul/ml tends to mislead in believing the essential oil as more active than it is. If instead of microliters miligrams were used, it would become obvious that the MIC against many species is not sufficiently low. For instance the 0.16 µL/mL, which seems very low, if converted to mg, assuming a density for the essential oil of 0.85, would be 160 *0.85=136 ug/ml, which is higher than the 100 ug/ml for very active extracts (“ to be considered a promising activity, a crude extract must demonstrate a MIC under 100 μg/mL”). (Bueno J. In Vitro Antimicrobial Activity of Natural Products Using Minimum Inhibitory Concentrations: Looking for New Chemical Entities or Predicting Clinical Response. Med. Aromat. Plants. 2012;1 doi: 10.4172/2167-0412.1000113). The same is true for most, if not all concentrations used throughout the paper. The authors should therefore preferably convert concentrations to ug/ml using the measured density of the oil or, alternatively, discuss critically these aspects in the paper.

Lines 167-168: We agree on the apparent superiority over fluconazole, but this is an in vitro study, not taking into account any PK considerations, because in vivo things may be different (in the context of systemic administration), and this is another limitation that should be discussed.

Figure 3: the magnifying power is not visible due to the low resolution of the image, one should be able to clearly see the magnification.

English language and style are generally fine, but in certain places small errors need correction.

Author Response

Reviewer #3

The paper is interesting and worth publishing, but a number of corrections have to be made, as discussed below.

  1. Line 2: Please use the complete name in the title, “Bupleurum rigidum subsp. paniculatum (Brot.) H.Wolff”. This should also be used for the first time in the full text and abstract.

The complete name of the species was included in the title and when first mentioned in the abstract and full text.

  1. Line 21: please correct the name to its complete form, as above.

This correction was done.

  1. Line 29: it is probably the effect “of the oil”, not “on the oil”.

This mistake was corrected.

  1. Line 32: please correct the name to “Bupleurum rigidum subsp. paniculatum”.

The correction was made.

  1. Lines 59-60: “undesirable side effects” are actually “safety issues”, therefore one of the two is sufficient here.

The words ‘undersirable side effects’ were removed from the sentence.

  1. Line 113: the meaning of “antivirulent” is confusing, therefore we suggest replacing it with a clearer term. The same holds true for other occurrences of the term.

To avoid confusions the term ‘ antivirulent’ was removed and the sentence ‘A combined antifungal and anti-inflammatory potential for B. rigidum subsp. paniculatum essential oil is reported, with emphasis on its antivirulent effect against C. albicans infections’ was altered to:

‘A combined antifungal and anti-inflammatory potential for B. rigidum subsp. paniculatum essential oil is reported, with emphasis on its effect against C. albicans virulence factors’

  1. Lines 508: the DMSO concentration should be stated.

The concentration of DMSO (1%) was included.

  1. Lines 590: for good reasons discussed in (https://journals.sagepub.com/doi/full/10.1177/0253717620933419) I would recommend replacing SEM with standard deviation.

The graphs were corrected and SEM values was replaced by SD.

  1. Line 118: the 0.6% yield was identical for the two samples? If so, this should be stated clearly, so as to avoid confusions.

The essential oil from Coimbra region presented a yield of 0.7% and the sample from Fátima showed a yield of 0.6%. This information was included in the results section.

  1. Lines 139-141: the fact that MICs are expressed as ul/ml tends to mislead in believing the essential oil as more active than it is. If instead of microliters miligrams were used, it would become obvious that the MIC against many species is not sufficiently low. For instance the 0.16 µL/mL, which seems very low, if converted to mg, assuming a density for the essential oil of 0.85, would be 160 *0.85=136 ug/ml, which is higher than the 100 ug/ml for very active extracts (“ to be considered a promising activity, a crude extract must demonstrate a MIC under 100 μg/mL”). (Bueno J. In Vitro Antimicrobial Activity of Natural Products Using Minimum Inhibitory Concentrations: Looking for New Chemical Entities or Predicting Clinical Response. Med. Aromat. Plants. 2012;1 doi: 10.4172/2167-0412.1000113). The same is true for most, if not all concentrations used throughout the paper. The authors should therefore preferably convert concentrations to ug/ml using the measured density of the oil or, alternatively, discuss critically these aspects in the paper.

As reviewer #2 also suggested to change the units, we have modified the MIC and MLC units to ug/mL. The oil has a density of 0.9 and this value was considered for the conversions.

  1. Lines 167-168: We agree on the apparent superiority over fluconazole, but this is an in vitro study, not taking into account any PK considerations, because in vivo things may be different (in the context of systemic administration), and this is another limitation that should be discussed.

We totally agree and the following sentence was included in the conclusion section.

Further in vivo studies are needed to validate the efficacy of the essential oil. Moreover, the clinical relevance of our results depends on the availability of the essential oil on target organs. Therefore, pharmacokinetic studies are essential to link in vitro effects to in vivo efficacy.

  1. Figure 3: the magnifying power is not visible due to the low resolution of the image, one should be able to clearly see the magnification.

The electron microscope images presents a good resolution but the scale bars indeed has a very low resolution and therefore figures were redone.

  1. English language and style are generally fine, but in certain places small errors need correction.

Detected errors were corrected.

Reviewer 4 Report

In the present manuscript, the antifungal and anti-inflammatory potential of Bupleurum subsp. paniculatum essential oil from Portugal were evaluated.

Lines 91-92: correct the meaning and use either the species or the genus in both sentences.

Line 99: add a coma after the first brackets.

Table 1: add SE or SD values in percentages of oil compounds. Correct comma in allo-Ocimene content of Coimbra sample.

Lines 131-132: the subheading should not refer directly to results.

Table 2: where these results refer to? Coimbra or Fatima sample? What were the positive controls? Define MIC and MLC in the footnote.

Figure 1b: why the % of filamentous yeasts is above 100% for 0.2μL/mL of fluconazole? The same question applies for Figure 2a (secon from last column).

Fluconazole MIC values should be presented in Table 2.

Figure 2: it is not clear what the error bars above the column represent. In Figure 2a there are two bars with one asterisk and in Figure 2b there are tow bars with one asterisk. Use Latin letters instead. The same applies for Figure 5.

Moreover, it is not clear which sample was used for the bioactive assays.

Line 440 and 453: correct the numbering of subheadings.

Author Response

Reviewer #4

In the present manuscript, the antifungal and anti-inflammatory potential of Bupleurum subsp. paniculatum essential oil from Portugal were evaluated.

  1. Lines 91-92: correct the meaning and use either the species or the genus in both sentences.

The sentence was corrected as it refers to the genera and not the species.

  1. Line 99: add a coma after the first brackets.

The comma was added.

  1. Table 1: Correct comma in allo-Ocimene content of Coimbra sample.

The comma was corrected.

  1. Lines 131-132: the subheading should not refer directly to results.

The subheading was modified to ‘The essential oil showed a broad-spectrum fungicidal effect’

  1. Table 2: where these results refer to? Coimbra or Fatima sample? What were the positive controls? Define MIC and MLC in the footnote.

The sample used to perform the bioactive assays was that from Coimbra region. This information was now clearly stated in the results section, before Table 1. In these assays, the positive control is inoculated medium with DMSO since it is used to dilute the essential oils. This information regarding negative and positive controls is stated in the methods section: ‘Serial dilutions of the oil (up to 0.036 µg/mL) were tested as well as a negative control (medium without fungi) and a positive control (inoculated medium with DMSO).

  1. Figure 1b: why the % of filamentous yeasts is above 100% for 0.2μL/mL of fluconazole? The same question applies for Figure 2a (secon from last column).

Fluconazole at 0.2μL/mL does not inhibit Candida albicans germ tube formation nor biofilm mass or viability. The slight increase above the control may be explained by some variability between experiments as seen by the standard deviation or the compound may alter the metabolic activity of the yeast.

  1. Fluconazole MIC values should be presented in Table 2.

The MIC and MLC values for fluconazole were included in table 2.

  1. Figure 2: it is not clear what the error bars above the column represent. In Figure 2a there are two bars with one asterisk and in Figure 2b there are two bars with one asterisk. Use Latin letters instead. The same applies for Figure 5.

For a better understanding, the statistical differences (asterisks and cardinals) were placed above the column and a more clear explanation on what the differences refer to was provided in the legend of the figures.

  1. Moreover, it is not clear which sample was used for the bioactive assays.

The sample from Coimbra was used for the bioactive assays. This information was stated in a more clear way in the results section (before table 1).

  1. Line 440 and 453: correct the numbering of subheadings.

The numbering was corrected.

Round 2

Reviewer 4 Report

The authors have addressed the suggested issues. Therefore, I recommend the acceptance of the manuscript in its present form.